# LoWR: LoRA Weight Rescaling for Effective Rank Utilization beyond Reduction

## Abstract

As the scale of large language models (LLMs) continues to increase, the parameter-efficient finetuning (PEFT) of LLMs has drawn much attention. One of the most popular PEFT methods is low-rank adaptation (LoRA), which has attracted numerous subsequent works aimed at improving it. While rank is one of the essential hyperparameters for LoRA, many previous works propose dynamically allocating rank to reduce the computational demand, *e.g.*, pruning insignificant channels to reduce the rank. This paper proposes a principled method to proactively guide the LoRA module to utilize the allocated rank fully. We first provide a new perspective to understand the difference between LoRA and full finetuning. We demonstrate that the two weight matrices in the LoRA module serve as proxies for the input and output gradients. Since the input of each layer is generally more stable than the gradient, the channel difference mainly reflects on the weight matrix at the left (weight matrix $B$). We further propose a principled plug-in method, grounded in theoretical analysis and empirical findings. Our proposed method reweights the two weight matrices in LoRA using a simple yet effective algorithm to further stabilize and encourage the training of insignificant channels. Experiments are conducted on widely used models (Llama, Mistral, *etc.*) and benchmarks (GSM8k, GLUE, SQuAD, *etc.*), where our proposed method significantly boosts the performance of LoRA and its variants, which suggests its universal and orthogonal effectiveness as a plug-in.

## 1 Introduction

Large Language Models (LLMs) have shown astonishing ability in solving various tasks [27; 8; 2]. Along with the increase in their performance, the number of parameters and the computational demands of LLMs pose significant challenges. To deal with the challenge and more efficiently adapt pre-trained LLMs, various parameter-efficient finetuning (PEFT) methods have been proposed [13; 1; 14; 16; 18]. Low-rank adaptation (LoRA) [14] is one of the most widely adopted techniques. Instead of updating the original weight matrix, LoRA reduces the number of parameters by updating two smaller weight matrices, whose product yields a low-rank weight matrix of the same shape. The efficiency and broad application of LoRA have attracted significant research interest in improving LoRA. Various methods have been proposed in seeking a better initialization [21; 3], better training hyper-parameters [10; 22; 24], *etc.* Notably, the rank as the key hyperparameter for LoRA controlling the balance between performance and efficiency has also received much attention. Rank allocation methods [26; 33; 20] propose to optimize the rank allocation for better performance and efficiency. However, most previous works passively treat the rank, *e.g.*, pruning the most insignificant channel to reduce the rank. In this paper, we propose investigating the role of the two weight matrices in LoRA from a channel perspective through theoretical analyses and proactively training the LoRA module to utilize the allocated rank fully with a principled method.

Generally, a LoRA module of rank $n$ has $n$ channels, each corresponding to a row vector in LoRA-A and a column vector in LoRA-B whose product is a weight matrix of rank 1.[1] The significance of each channel corresponds to the only non-zero singular value of the rank 1 matrix, *i.e.* the product of the $l_2$ norm of the two weight vectors, which we call the significance of the channel. To analyze the causes of the difference between channels in a LoRA module, we first provide a new perspective on

---

[1]Typically, the matrix on the right is named LoRA-A, and the matrix on the left is named LoRA-B.

understanding the difference between LoRA and full finetuning. In full finetuning, the gradient of the weight matrix is the product of the input and the output gradient. Compared to full finetuning, we show that the two weight vectors of each channel act as proxies for the input and the output gradient, respectively. By updating along the gradient of the two matrices LoRA-A and LoRA-B separately, the full-weight matrix is updated both by the product of the row vectors in LoRA-A with the output gradient and the product of the column vectors in LoRA-B with the input. Unlike previous works that treat the two weight matrices LoRA-A and LoRA-B as the left and right singular matrices of the full weight matrix, our analysis reveals their relationship with the input and output gradients and their role during LoRA optimization.

From our new perspective, we further analyze the difference between LoRA channels. As shown in [9; 7], the gradient of deep neural networks could vary even at the late stage of training. Therefore, the input of each layer is generally more stable than the output gradient during finetuning. As a result, we empirically show that the difference mainly comes from the weight matrix on the left in LoRA (namely LoRA-B in most cases) acting as the proxy of the output gradient. Based on this phenomenon, we propose a simple and effective plug-in method that applies channel-wise adjustment, enlarging LoRA-A and reducing LoRA-B. The resulting benefit is twofold. Firstly, it emphasizes the role of LoRA-A as the proxy of input, which is more stable. Secondly, it fastens the update on LoRA-B, encouraging full use of each channel. Experiments are conducted on widely used models (Llama, Mistral, *etc.*) and benchmarks (GSM8k, GLUE, and so on), where our method effectively boosts the performance of different variants of LoRA as a plug-in. **The contributions are:**

- We provide a new perspective on understanding the difference between LoRA and full finetuning. To the best of our knowledge, we are the first to demonstrate that the two weight matrices in LoRA serve as proxies for the input and the output gradient, which differs from previous works that treat the two weight matrices in LoRA as a decomposition of the full weight matrix.
- Based on the fact that the input is generally more stable than the output gradient, we empirically show that the difference in weight norms between channels mainly comes from LoRA-B (the matrix on the right acting as the output gradient proxy), providing insights for a better understanding of the optimization of LoRA.
- Based on our theoretical analysis and empirical findings, we propose a principled method to actively train the LoRA module to use the allocated rank fully, instead of passively pruning insignificant channels. Experimental results are provided to verify the effectiveness of our method.

**Remark.** Perhaps the most related work to ours is LoRA+ [], which can also serve as a plug-in across LORA and its variants. Experimentally, we show that our method can be combined with LoRA+ to boost each other. Technically, we provide a more fine-grained understanding and mechanism at the channel level.

## 2 PRELIMINARY AND RELATED WORKS

Low-rank Adaptation (LoRA) [14] is a method proposed to reduce the number of parameters during finetuning. For the weight matrix of a linear layer $\mathbf{W} \in \mathbb{R}^{m \times n}$, instead of directly updating the weight matrix, LoRA proposes to update two smaller matrices $\mathbf{A} \in \mathbb{R}^{r \times n}$ and $\mathbf{B} \in \mathbb{R}^{m \times r}$ and update the weight by adding the product of the two small matrices to a fixed weight matrix:

$$\mathbf{W}_{lora} = \mathbf{W} + \mathbf{BA}. \tag{1}$$

Changing the rank $r$ allows one to adjust the number of parameters and balance efficiency and performance. Due to its efficacy, LoRA has become one of the most popular PEFT methods, attracting various works to improve it. A branch of methods focuses on the initialization of LoRA, where the original method initializes $\mathbf{A}$ with Kaiming initialization [11] and initializes $\mathbf{B}$ with a zero matrix. To improve the initialization, Meng et al. [21] proposes to initialize the two matrices based on the singular value decomposition (SVD) of the pre-trained weight, and Wang et al. [30] proposes to initialize LoRA based on the gradient of the pre-trained weight. Some methods focus on improving the optimization of LoRA, *e.g.* [3; 29] propose to regularize the channels in the LoRA module to be orthogonal to each other, [20; 31] propose to optimize the direction and magnitude of channels in LoRA separately. This community also witnessed many methods in dynamically allocating the rank of LoRA. [33] proposes to keep the LoRA in an SVD-like format where channels with small singular values are pruned during training. Meanwhile, [32; 19] proposes to start with rank 1 and gradually increase the rank based on an importance score on each module. In table 1, we present a

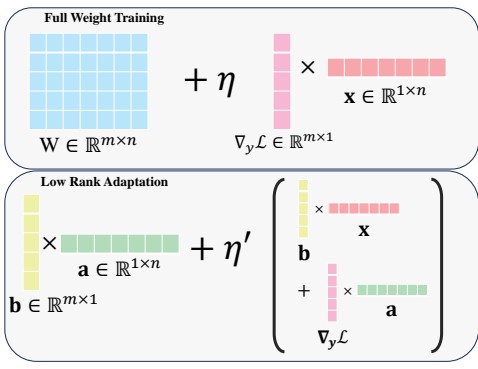

| Perspective | Description | Methods |
|---|---|---|
| Initialization | Initialize LoRA by decomposing the pre-trained weight or tracking the gradient | PiSSA [21], OLoRA [3], LoRA-GA [30], *etc.* |
| Optimization | Focus on the optimization of LoRA module, *e.g.*, restraining the channels to be orthogonal, adjusting the learning rate. | LoRA+ [10], HydraLoRA [24], BiLoRA [22],*etc.* |
| Rank Allocation | Allocate rank across layers or modules, *e.g.*, dynamically remove insignificant channels. | DyLoRA [26], AdaLoRA [33], DoRA [20], *etc.* |

Table 1: A simple summary of previous works on improving LoRA. Generally, previous works treat LoRA as weight decomposition. This paper provides a new perspective on understanding the two weight matrices in LoRA as the proxy of inputs and output gradients during finetuning. Based on our analysis, we propose a principled method to stabilize the optimization of LoRA further and lead the LoRA module to make full use of the allocated rank.

Figure 1: An illustration of the difference between full-weight update and low-rank adaptation (LoRA). We use a LoRA of rank $1$ to illustrate, where the update in the LoRA weight comprises two terms. The weight vector $\mathbf{b}$ acts as the proxy output gradient, and the weight vector $\mathbf{a}$ acts as the proxy input.

simple summary of previous works focusing on improving LoRA from different perspectives. While most previous works treat LoRA as a weight decomposition of the updated weight, we provide a new perspective on how LoRA works in this paper. We show that the two weight matrices are the proxy of input and output gradient during training, and the optimization of LoRA has a fundamental difference from full finetuning. Based on our analysis, we propose a new plug-in method to stabilize the optimization of LoRA and guide the LoRA module in using the allocated rank fully.

## 3 CLOSER LOOK ON LoRA: THEORETICAL INSIGHTS

### 3.1 PRELIMINARY: THE CHANNELS IN LoRA

As shown in Eq. 1, the LoRA updates the model by updating two weight matrices $\mathbf{A} \in \mathbb{R}^{r \times n}$ and $\mathbf{B} \in \mathbb{R}^{m \times r}$ where $r$ is the rank. There are $r$ channels in the LoRA module, where each channel corresponds to a row vector in $\mathbf{A}$ and a column vector in $\mathbf{B}$. For the $i$-th channel, we denote the two vector as $\mathbf{a}_i \in \mathbb{R}^n, \mathbf{b}_i \in \mathbb{R}^m$ where $\mathbf{A} = (\mathbf{a}_1, \mathbf{a}_2, \cdots, \mathbf{a}_r)^\top$ and $\mathbf{B} = (\mathbf{b}_1, \mathbf{b}_2, \cdots, \mathbf{b}_r)$. Therefore, we could rewrite Eq. 1 from the channel perspective:

$$\mathbf{W}_{lora} = \mathbf{W} + \sum_i^r \mathbf{b}_i \mathbf{a}_i^\top. \tag{2}$$

The LoRA weight $BA$ is actually the summation of $r$ matrices, each corresponding to a channel. As the product of two vectors, each matrix is of rank $1$ with one non-zero singular value $\|\mathbf{a}_i\| \cdot \|\mathbf{b}_i\|$. The non-zero singular value indicates the significance of the channel, *i.e.*, how much the channel contributes to the weight $W_{lora}$. Generally, a larger rank enables the LoRA module to learn more significant channels, which will result in better performance. In the low-rank regime, every channel counts; therefore, learning insignificant channels is undesirable. We have witnessed many previous rank allocation methods [33; 26], pruning insignificant channels to reduce rank. In these works, the two weight matrices are generally taken as a decomposition of the full weight matrix, *e.g.* the LoRA module is regularized to be similar to an SVD decomposition where the row vectors of $A$ (and the column vectors of $B$) are regularized to be orthogonal to each other. In this section, we provide a brand new perspective to understand the role of the two weight matrices $A$ and $B$.

### 3.2 REACQUAINT WITH THE TWO WEIGHT MATRICES IN LoRA

First consider a linear layer $\mathbf{y} = \mathbf{W}\mathbf{x}$ where $\mathbf{y} \in \mathbb{R}^m$ is the output, $\mathbf{x} \in \mathbb{R}^n$ is the input, and $W \in \mathbb{R}^{m \times n}$ is the weight. Suppose we have a loss $\mathcal{L}$, by the chain rule:

$$\nabla_{\mathbf{W}}\mathcal{L} = (\nabla_{\mathbf{y}}\mathcal{L})\mathbf{x}^\top, \tag{3}$$

where $\nabla_{\mathbf{W}}\mathcal{L} \in \mathbb{R}^{m \times n}$ is the weight gradient and $\nabla_{\mathbf{y}}\mathcal{L} \in \mathbb{R}^m$ is the gradient of the output.

When it comes to the LoRA, according to Eq. 2, we have,

$$\mathbf{y} = (\mathbf{W} + \sum_i^r \mathbf{b}_i \mathbf{a}_i^\top)\mathbf{x} \tag{4}$$

For $\mathbf{a}_i \in \mathbb{R}^n$, by the chain rule, we have,

$$\nabla_{\mathbf{a}_i}\mathcal{L} = \left(\mathbf{b}_i^\top \nabla_{\mathbf{y}}\mathcal{L}\right)\mathbf{x}. \tag{5}$$

It means that the gradient of $\mathbf{a}_i$ is in the direction of input $\mathbf{x}$ with its magnitude determined by the inner product between $\mathbf{b}_i$ and the output gradient $\nabla_{\mathbf{y}}\mathcal{L}$. Similarly, we have,

$$\nabla_{\mathbf{b}_i}\mathcal{L} = \left(\mathbf{a}_i^\top \mathbf{x}\right)\nabla_{\mathbf{y}}\mathcal{L}, \tag{6}$$

where the gradient of $\mathbf{b}_i$ is in the direction of the output gradient, with its magnitude determined by the inner product between $\mathbf{a}_i$ and the input $\mathbf{x}$.

Suppose we update the model along the direction of the gradient with a sufficiently small learning rate $\eta$. For full finetuning, we have,

$$\mathbf{W}' = \mathbf{W} + \eta\nabla_{\mathbf{W}}\mathcal{L} = W + \eta\left(\nabla_{\mathbf{y}}\mathcal{L}\right)\mathbf{x}^\top, \tag{7}$$

where $\mathbf{W}'$ represents the updated weight matrix. For LoRA, since the two matrices are separately updated, we have,

$$\mathbf{W}'_{lora} = \mathbf{W} + \sum_i^r (\mathbf{b}_i + \eta\nabla_{\mathbf{b}_i}\mathcal{L})(\mathbf{a}_i + \eta\nabla_{\mathbf{a}_i}\mathcal{L})^\top. \tag{8}$$

Derive Eq. 8, the weight $\mathbf{W}_{lora}$ is changed by:

$$\sum_i^r \left[\eta\left(\nabla_{\mathbf{b}_i}\mathcal{L}\right)\mathbf{a}_i^\top + \eta\mathbf{b}_i\left(\nabla_{\mathbf{a}_i}\mathcal{L}\right)^\top + \eta^2\left(\nabla_{\mathbf{b}_i}\mathcal{L}\right)\left(\nabla_{\mathbf{a}_i}\mathcal{L}\right)^\top\right] \tag{9}$$

.

Since $\eta$ is a small value, we mainly consider $\eta\left(\nabla_{\mathbf{b}_i}\mathcal{L}\right)\mathbf{a}_i^\top + \eta\mathbf{b}_i\left(\nabla_{\mathbf{a}_i}\mathcal{L}\right)^\top$ and ignore the rather small value $\eta^2\left(\nabla_{\mathbf{b}_i}\mathcal{L}\right)\left(\nabla_{\mathbf{a}_i}\mathcal{L}\right)^\top$. Combining Eq. 9 with Eq. 5 and Eq. 6, we further have,

$$\begin{aligned} \mathbf{W}'_{lora} - \mathbf{W}_{lora} &= \sum_i^r \left[\eta[(\mathbf{a}_i^\top \mathbf{x})\nabla_{\mathbf{y}}\mathcal{L}]\mathbf{a}_i^\top + \eta[(\mathbf{b}_i^\top \nabla_{\mathbf{y}}\mathcal{L})\mathbf{b}_i]\mathbf{x}^\top\right] \\ &= \eta\left[\sum_i^r (\mathbf{a}_i^\top \mathbf{x})\cdot[(\nabla_{\mathbf{y}}\mathcal{L})\mathbf{a}_i^\top] + \sum_i^r (\mathbf{b}_i^\top \nabla_{\mathbf{y}}\mathcal{L})\cdot[\mathbf{b}_i\mathbf{x}^\top]\right]. \end{aligned} \tag{10}$$

When comparing the change in LoRA weights as in Eq. 10 to the change of weight in full finetuning as in Eq. 7, it is easy to notice that each $\mathbf{a}_i$ is in the place of the input $\mathbf{x}$ and weighted by the inner product $\mathbf{a}_i^\top \mathbf{x}$ and each $\mathbf{b}_i$ is in the place of the output gradient $\nabla_{\mathbf{y}}\mathcal{L}$ weighted by $\mathbf{b}_i^\top \nabla_{\mathbf{y}}\mathcal{L}$.

**Proposition 3.1.** *[LoRA as proxy] During the training of a LoRA adapter, each row vector $\mathbf{a}_i$ in the weight matrix $\mathbf{A}$ is updated based on the direction of the input $\mathbf{x}$ and acts as a proxy of the input. Each column vector $\mathbf{b}_i$ in the weight matrix $\mathbf{B}$ is updated based on the direction of the output gradient $\nabla_{\mathbf{y}}\mathcal{L}$ and acts as a proxy of the output gradient.*

In Proposition 3.1, we provide a new perspective to understand LoRA, different from most previous works that view LoRA as a low-rank decomposition of the weight matrix and treat the two matrices $\mathbf{A}$ and $\mathbf{B}$ like the left and the right singular matrix. Instead, the matrix $A$ captures the direction of the input and acts as its proxy. The matrix $\mathbf{B}$ captures the direction of the output gradient and acts as its proxy. For each channel, the two corresponding weight vectors $\mathbf{a}_i$ and $\mathbf{b}_i$ are a pair of proxies for the input $\mathbf{x}$ and the output gradient $\nabla_{\mathbf{y}}\mathcal{L}$. These two vectors affect each other's gradient magnitude. The gradient of $\mathbf{b}_i$ would be larger when the input $\mathbf{x}$ is more similar to $\mathbf{a}_i$, and the same goes for $\mathbf{a}_i$.

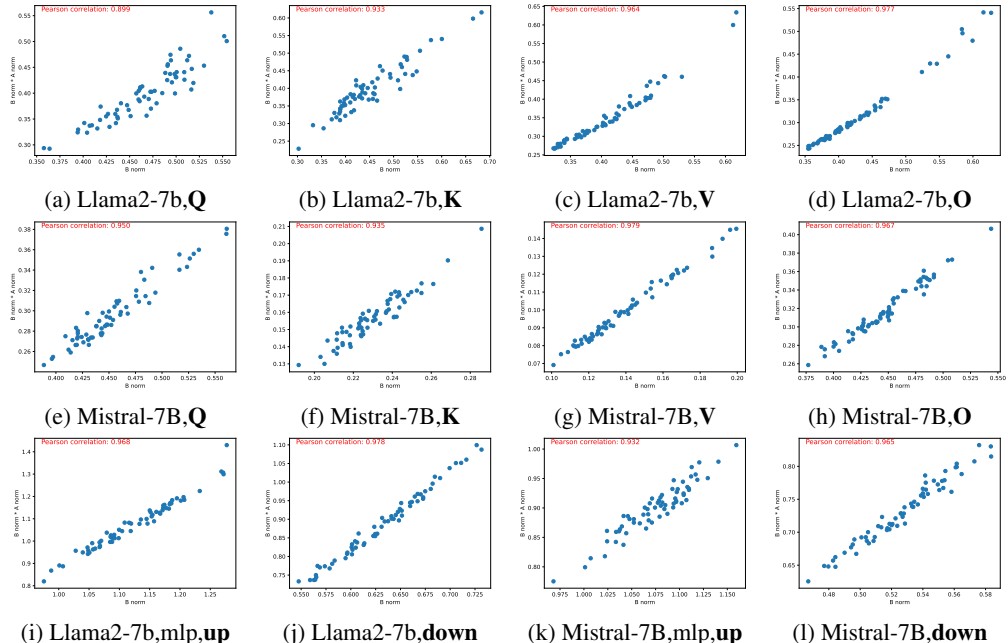

(a) Llama2-7b,**Q**     (b) Llama2-7b,**K**     (c) Llama2-7b,**V**     (d) Llama2-7b,**O**

(e) Mistral-7B,**Q**     (f) Mistral-7B,**K**     (g) Mistral-7B,**V**     (h) Mistral-7B,**O**

(i) Llama2-7b,mlp**up**     (j) Llama2-7b,**down**     (k) Mistral-7B,mlp**up**     (l) Mistral-7B,**down**

Figure 2: The relationship between $\|\mathbf{a}_i\| \cdot \|\mathbf{b}_i\|$ and $\|\mathbf{b}_i\|$, where the y-axis corresponds to $\|\mathbf{a}_i\| \cdot \|\mathbf{b}_i\|$ and the x-axis corresponds to $\|\mathbf{b}_i\|$. Each dot represents a channel in the LoRA Module. We present results of different LoRA modules finetuned on GSM8K [5] at the first layer of Llama-2-7b [25] and Mistral-7B-v0.1 [15]. We provide Pearson's correlation coefficient at the top-left of each figure. Generally, the difference in $\|\mathbf{a}_i\| \cdot \|\mathbf{b}_i\|$ mainly reflects on $\|\mathbf{b}_i\|$ with Pearson's r larger than 0.9. For more results, please refer to Appendix A.

## 3.3 INSIGHTS ON THE CHANNEL DIFFERENCE

As introduced in Sec. 3.2, we propose a new perspective to understand LoRA, where each channel in LoRA corresponds to a pair of proxies for the input and the output gradient. From this perspective, we rethink the rank in the LoRA module, the key hyperparameter that controls the number of proxies the adapter would have for the input and the output gradient. As shown in [4], the Matthew effect could exist between similar channels, *i.e.* the channels with larger weight whose weight would be updated faster since the larger weight in one vector induces a larger gradient on its counterpart. This further poses the question of how we can help with the update and guide the module in using the rank fully. To address this question, we further analyze the difference between channels.

As the name "finetune" suggests, the model does not change rapidly during finetuning. Therefore, we would expect a relatively stable input for each layer. On the contrary, as shown by previous works [9], the gradient might change drastically even close to convergence. In terms of the loss landscape, previous works [6; 7] indicate that the loss landscape's sharpness may even increase as the model becomes closer to the minimum. Since $\mathbf{a}_i$ is the proxy for the input and $\mathbf{b}_i$ is the proxy for the output gradient, we expect the difference in the significance of each channel, *i.e.* the product of the norm of the two vectors $\|\mathbf{a}_i\|\|\mathbf{b}_i\|$, mainly reflect on the difference in the $\|\mathbf{b}_i\|$.

To empirically verify our analysis, we collect the $l_2$ norm of each row vector in the matrix $\mathbf{A}$ and each column vector in the matrix $\mathbf{B}$ of LoRA modules for the widely used Llama2-7b [25] and Mistral-7b [15] finetuned on GSM8K [5], one of the most popular math benchmark. In Fig. 2, we demonstrate the result where the y-axis corresponds to $\|a_i\| \cdot \|b_i\|$ and the x-axis corresponds to $\|b_i\|$. Each dot corresponds to a channel in the LoRA module. Due to the page limit, we provide the results of the LoRA modules in the first layer. Please refer to Appendix A for more results. Generally, the significance of each channel (*i.e.* $\|a_i\| \cdot \|b_i\|$) is positively correlated with the difference in $\|b_i\|$ with Pearson's r close to or greater than 0.9 for all the modules. This phenomenon verifies our analysis that the difference between channels mainly reflects on $b_i$, the column vectors in $\mathbf{B}$. Since the gradient can vary drastically, the vector $b_i$ for those insignificant channels may end up learning noise in the output gradient and fail to serve as a proxy for the output gradient. In the next section, we further provide our analysis of the LoRA optimization and a plug-in to improve LoRA.

# 4 A PLUG-IN METHOD TO IMPROVE LoRA

In this section, we provide an analysis of the differences between LoRA and full finetuning in terms of optimization. We demonstrate that even two equivalent LoRA modules can exhibit significantly different optimization dynamics. Based on our analysis, we propose a plug-in method to improve LoRA. Extensive experiments are conducted to verify the effectiveness of our proposed method.

## 4.1 LoWR: LoRA WEIGHT RESCALING BETWEEN $\mathbf{A}$ AND $\mathbf{B}$

As analyzed in Sec. 3, the $\mathbf{A}$ and $\mathbf{B}$ in LoRA play different roles (*i.e.* the proxy of the input and the proxy of the output gradient). According to Eq. 10, when we update the model with the gradient and a sufficiently small learning rate $\eta$, the equivalent gradient on $\mathbf{W}_{lora}$ could be approximated by

$$\sum_i^r (\mathbf{a}_i^\top \mathbf{x}) \cdot [(\nabla_{\mathbf{y}}\mathcal{L})\mathbf{a}_i^\top] + \sum_i^r (\mathbf{b}_i^\top \nabla_{\mathbf{y}}\mathcal{L}) \cdot [\mathbf{b}_i \mathbf{x}^\top]. \tag{11}$$

**Proposition 4.1.** *[Optimization Variety] Even when all the conditions are identical, two equivalent LoRA modules can end up with totally different optimization trajectories.*

Proposition 4.1 indicates the difference and the complexity in optimization brought by LoRA. The equivalent gradient[2] of $\mathbf{W}_{lora}$ as shown in Eq. 11 could vary even though $\mathbf{W}_{lora}$ does not change and all the conditions are identical. We provide a proof sketch in the following.

*Proof.* For a LoRA module $\mathbf{W}_{lora} = \mathbf{W} + \mathbf{BA}$, we can have an equivalent module $\mathbf{W}'_{lora}$ with $\mathbf{A}' = n\mathbf{A}$ and $\mathbf{B}' = \frac{1}{n}\mathbf{B}$ where $n \neq 1$. When we update the two LoRA modules, the updated weights are different:

$$n^2 \sum_i^r (\mathbf{a}_i^\top \mathbf{x}) \cdot [(\nabla_{\mathbf{y}}\mathcal{L})\mathbf{a}_i^\top] + \frac{1}{n^2} \sum_i^r (\mathbf{b}_i^\top (\nabla_{\mathbf{y}}\mathcal{L})) \cdot [\mathbf{b}_i \mathbf{x}^\top]$$

$$\neq \sum_i^r (\mathbf{a}_i^\top \mathbf{x}) \cdot [(\nabla_{\mathbf{y}}\mathcal{L})\mathbf{a}_i^\top] + \sum_i^r (\mathbf{b}_i^\top (\nabla_{\mathbf{y}}\mathcal{L})) \cdot [\mathbf{b}_i \mathbf{x}^\top].$$

$\square$

To further stabilize the optimization of the LoRA module and guide it to learn more significant channels, we propose rescaling the two weight matrices $\mathbf{A}$ and $\mathbf{B}$. Note that the norm $\|\mathbf{a}_i\|$ and $\|\mathbf{b}_i\|$ controls the weight for the two parts $(\mathbf{a}_i^\top \mathbf{x}) \cdot (\nabla_{\mathbf{y}}\mathcal{L})\mathbf{a}_i^\top$ and $(\mathbf{b}_i^\top \nabla_{\mathbf{y}}\mathcal{L}) \cdot [\mathbf{b}_i \mathbf{x}^\top]$ in Eq. 11. By rescaling $\mathbf{a}_i$ and $\mathbf{b}_i$, one can adjust the proportions of the two parts in the updated weight and thus change the equivalent gradient without changing the equivalent weight $\mathbf{W}_{lora}$.

According to our analysis and empirical results in Sec. 3.3, $\mathbf{a}_i$ is more stable as the proxy of the input while $\mathbf{b}_i$ is rather unstable to perform as the proxy of the output gradient and might learn noises of the output gradient. A straightforward choice to stabilize the optimization would be emphasizing $(\mathbf{a}_i^\top \mathbf{x}) \cdot [(\nabla_{\mathbf{y}}\mathcal{L})\mathbf{a}_i^\top]$ by enlarging $\|\mathbf{a}_i\|$. By increasing the ratio $\|\mathbf{a}_i\|/\|\mathbf{b}_i\|$, the benefits are twofold:

- Enlarge $(\mathbf{a}_i^\top \mathbf{x}) \cdot [(\nabla_{\mathbf{y}}\mathcal{L})\mathbf{a}_i^\top]$ and reduce $(\mathbf{b}_i^\top \nabla_{\mathbf{y}}\mathcal{L}) \cdot [\mathbf{b}_i \mathbf{x}^\top]$ in Eq. 11. As the proxy of the input, $\mathbf{a}_i$ is more stable than $\mathbf{b}_i$ as the proxy of the output gradient.
- Enlarge the $\nabla_{\mathbf{b}_i}\mathcal{L}$ as shown in Eq. 6. It would accelerate the updating of $\mathbf{b}_i$, helping $\mathbf{b}_i$ in insignificant channels to learn the gradient direction more effectively.

Notably, previous work [10] shows that, for the gradient to be $O(1)$, the learning rate for $\mathbf{b}$ should be larger than that for $\mathbf{a}$. From a different perspective, we also demonstrate that not only should the $\mathbf{b}$ be updated faster, but the norm of $\mathbf{a}$ should be amplified since $\mathbf{a}$ is a more stable proxy for input $\mathbf{x}$. In terms of adjusting the training dynamics of a LoRA module, rescaling the two matrices achieves a similar effect as adjusting the learning rate as proposed in LoRA+ [10]. However, since our method is proposed to balance between channels of the LoRA module, by rescaling each channel, we conduct a more fine-grained adjustment to proactively affect the training dynamics of the LoRA module during training and encourage the update on insignificant channels. Due to the limited space, we provide a detailed analysis of the connection and differences between our method and LoRA+ in Appendix A, along with experimental results combining our method with LoRA+.

---

[2]The gradient for $\mathbf{W}_{lora}$ is $\frac{\partial \mathcal{L}}{\partial \mathbf{y}}\mathbf{x}^\top$, the same in Eq. 3. We use the term "equivalent gradient" to indicate that $\mathbf{W}_{lora}$ is updated by the term in Eq. 11 with learning rate $\eta$.

Table 2: Experimental results on GSM8K [5] (test accuracy %). We report the result of LoRA [14] with or without our proposed method LoWR for Llama2-7b [25] and Mistral-7B-v0.1 [15].

| Model | Llama-2-7b | | | | Mistral-7B-v0.1 | | | |
|---|---|---|---|---|---|---|---|---|
| Rank | 16 | | 64 | | 16 | | 64 | |
| Weight type | float | 4bit | float | 4bit | float | 4bit | float | 4bit |
| vanilla LoRA [14] | 39.50% | 38.21% | 40.18% | 39.20% | 54.13% | 52.69% | 56.10% | 54.51% |
| PiSSA [21] | 40.56% | 38.06% | 39.27% | 38.06% | 54.21% | 53.45% | 55.88% | 54.66% |
| OLoRA [3] | 39.04% | 40.18% | 40.26% | 39.55% | 54.66% | 53.30% | 56.41% | 55.42% |
| vanilla w/ LoWR | 40.18% | 40.56% | **43.21**% | 39.42% | 54.89% | 54.36% | 56.33% | 55.04% |
| PiSSA w/ LoWR | **40.71**% | 39.12% | 41.47% | 39.42% | 54.97% | 54.97% | 55.80% | 54.66% |
| OLoRA w/ LoWR | 39.35% | **40.86**% | 39.88% | **39.65**% | **55.65**% | **56.71**% | **56.89**% | **55.95**% |

To stabilize the training of LoRA module, the ratio $\|\mathbf{a}_i\|/\|\mathbf{b}_i\|$ should be bounded. Otherwise, the $\mathbf{a}_i$ can not be sufficiently trained with the gradient $\nabla_{\mathbf{a}_i}\mathcal{L}$ is affected by $\mathbf{b}_i^\top \nabla_{\mathbf{y}}\mathcal{L}$ as in Eq. 5. The training of $\mathbf{b}_i$ may also fail to converge with a large gradient.

The above analyses lead to our simple plug-in method, LoRA weight Reweighting (LoWR). For each channel, we propose to rescale between the two weight vectors $\mathbf{a}_i$ and $\mathbf{b}_i$ during training. When $\|\mathbf{a}_i\|/\|\mathbf{b}_i\|$ is below a certain threshold, We multiply $\mathbf{a}_i$ with certain ratio $\gamma$ and divide $\mathbf{b}_i$ by the ratio $\gamma$. The ratio $\gamma$ is greater than 1, determined by

$$\gamma = 1 + \alpha \cdot [\max(\|\mathbf{b}\|) - \|\mathbf{b}_i\|] \cdot \sigma^2(\|\mathbf{a}\|\|\mathbf{b}\|), \tag{12}$$

where $\alpha$ is a hyper-parameter, $\max(\|\mathbf{b}\|)$ represent the maximum norm of the column vectors in $B$ and $\sigma^2(\|\mathbf{a}\|\|\mathbf{b}\|)$ is the variance of the product $\|\mathbf{a}_i\| \cdot \|\mathbf{b}_i\|$ between channels.

---

**Algorithm 1** LoRA Weight Rescaling (LoWR)

**Input:** Trainable module $f(\cdot)$, fine-tune dataset $D$, loss function $\mathcal{L}(\cdot, \cdot)$ threshold $\tau$, ratio $\alpha$.
**Output:** Fine-tuned module $f(\cdot)$
1: **for** $d$ in $D$ **do**
2:    update $f(\cdot)$ with $\mathcal{L}(f, d)$;
3:    **for** $i = 1, 2, \cdots, r$ **do**
4:       **if** $\frac{\|\mathbf{a}_i\|}{\|\mathbf{b}_i\|} < \tau$ **then**
5:          $r \leftarrow \alpha[\max(\|\mathbf{b}\|) - \|\mathbf{b}_i\|] \cdot \sigma^2(\|\mathbf{a}\|\|\mathbf{b}\|)$
6:          $\gamma \leftarrow 1 + r$;
         ▷ Determine the ratio for re-scaling $\mathbf{a}_i$ and $\mathbf{b}_i$
7:          $\mathbf{a}_i \leftarrow \gamma \cdot \mathbf{a}_i$;
8:          $\mathbf{b}_i \leftarrow \frac{1}{\gamma} \cdot \mathbf{b}_i$;
9:       **end if**
10:    **end for**
11: **end for**
12: **return** $f(\cdot)$

---

The method is proposed to increase the gradient on smaller $\mathbf{b}$ vectors. To achieve this goal, we determine the rescaling ratio with the term $\max(\|\mathbf{b}\|) - \|\mathbf{b}_i\|$, which aims to increase the rescaling ratio for those channels with a smaller $\mathbf{b}$, and the term $\sigma^2(\|\mathbf{a}\|\|\mathbf{b}\|)$, enlarging the rescaling ratio when the channels are more imbalanced. By rescaling $\mathbf{b}_i$ with $\frac{1}{\gamma}$ and $\mathbf{a}_i$ with $\gamma$, without changing the equivalent model, we expect to further stabilize and improve the optimization for LoRA, especially for those insignificant channels. The detailed algorithm is in Algorithm 1.

### 4.2 EXPERIMENTS

Following the previous works on PEFT, we conduct experiments on benchmarks (*e.g.* GSM8K [5], GLUE [28], *etc.*) with widely used models (such as Llama [25], mistral [15], *etc.*), where our method works as a plug-in on LoRA [14] and its variants [21; 3]. In particular, we further provide empirical results showing how LoWR affects the optimization of LoRA and mitigates insignificant channels as analyzed in Sec. 3.

#### 4.2.1 BENCHMARKING LoWR AS A PLUG-IN FOR LoRA

We follow the standard setting in previous works [17] to conduct experiments on widely used benchmarks such as GSM8K with Llama-2-7b and Mistral-7B-v0.1. Our proposed method, LoWR, has demonstrated its effectiveness in improving LoRA and its variants. In the following, we introduce experimental results on each benchmark. For more details on the experimental setting and hyperparameters, please refer to Appendix B.

**Experiments on GSM8K.** We conduct experiments on GSM8K following the setting in [17]. GSM8K (Grade School Math 8K) is a dataset of 8,500 high-quality, linguistically diverse grade school math problems. The dataset includes problems that take between 2 and 8 steps to solve, primarily involving basic arithmetic operations. We apply LoWR to vanilla LoRA and two variants

Table 3: The results of DeBERTa-V3-base [12] on GLUE datasets. We use the default hyperparameter setting in LoftQ to finetune the model on all nine tasks. Generally, combining LoftQ with our plug-in LoWR could boost the performance.

| Method | Rank | MNLI Acc | QNLI Acc | RTE Acc | SST Acc | MRPC Acc | CoLA Matt | QQP Acc | STSB Pearson | WNLI Acc |
|---|---|---|---|---|---|---|---|---|---|---|
| Full FT* | - | 90.5 | 94.0 | 82.0 | 95.3 | 89.5 | 69.2 | 92.4 | 91.6 | 59.8 |
| LoftQ | 32 | 87.64 | 91.58 | 58.12 | 92.0 | 77.45 | 52.7 | 91.0 | 88.70 | 49.0 |
| LoftQ+LoWR | 32 | 87.64 | **92.15** | **59.21** | **94.84** | **78.43** | **53.54** | **91.53** | **88.88** | **56.34** |

(PiSSA [21], OLoRA [3]). The two variants initialize LoRA by decomposing pre-trained weights and using the most significant components to initialize the LoRA module. PiSSA uses SVD decomposition, and OLoRA uses QR decomposition.

We report the results in Table 2 where the best results are in bold. We conduct experiments on Llama-2-7b and Mistral-7B-v0.1. The hyper-parameter for LoWR is grid-searched within a very small set ($\tau \in \{+\infty, 10, 100\}$, $\alpha \in \{0.1, 1\}$). With LoWR, the performance of the LoRA modules on GSM8K is significantly improved.

**Experiments on GLUE.** We conduct experiments on GLUE [28] with LoftQ [17], a method for better quantized LoRA initialization. The General Language Understanding Evaluation (GLUE) dataset is a widely used benchmark suite. It consists of nine sentence- or sentence-pair language understanding tasks. For example, the Stanford Sentiment Treebank (SST-2) task in GLUE focuses on binary sentiment analysis, determining whether a given sentence expresses positive or negative sentiment. We finetune DeBERTa-V3-base [12] with LoftQ using uniform quantization with 2-bit precision following the default setting in LoftQ. We report the results in Table 3. We use the default parameter setting for all the tasks and apply grid-search on hyperparameters $\tau$ and $\alpha$ in Algorithm 1 in a small set($\tau \in \{+\infty, 10, 100\}$, $\alpha \in \{0.1, 1\}$). Since we use the default setting for all tasks in GLUE, the performance of the vanilla LoftQ is worse than that reported in its original paper. "*" indicates the result is adopted from the LoftQ paper [17]. The results show the effectiveness of LoWR as a plug-in for LoftQ. Note that by limiting the grid search of hyperparameters to a small set, we want to demonstrate the robustness of the hyperparameter setting for our method. Generally, with smaller $\tau$ and $\alpha$, our method makes less change to the training dynamics of LoRA modules. Therefore, in a new scenario, one could steadily increase $\tau$ and $\alpha$ for a better performance. In Appendix A, we further provide an analysis on the sensitivity of LoWR to hyper parameters.

**Combining Our Method with AdaLoRA.**
While our method aims at changing the training dynamics of LoRA modules to make full use of the allocated rank, it is in fact orthogonal to those adaptive rank allocation methods in allocating different rank to different LoRA modules. Following AdaLoRA [33], we finetune DeBERTa-V3-base [12] on SQuAD [23]. Stanford Question Answering Dataset (SQuAD) is a reading comprehension dataset. We use the de-

Table 4: Results of fintuning DeBERTa-V3 [12] on SQuAD [23].

| Methods | Exact match | F1 |
|---|---|---|
| AdaLoRA [33] | 87.69 | 93.68 |
| AdaLoRA + LoWR | **87.92** | **93.81** |

fault hyperparameter in AdaLoRA [33] to finetune the model. As shown in Table 4, we present the exact match and F1 score of the finetuned model. The results show that our method could improve the performance of AdaLoRA as a plug-in.

### 4.2.2 THE CHANGE IN CHANNEL SIGNIFICANCE

In this section, we present the change in channel significance (the norm product $\|\mathbf{a}_i\| \cdot \|\mathbf{b}_i\|$. By LoWR, we want to accelerate the updating on $\mathbf{b}_i$ and stabilize the training by, especially for insignificant channels. In Fig. 3a, we present the scatter of channel norms comparing the vanilla LoRA with or without LoWR. The result is the LoRA module for the q_proj in the first layer of Llama-2-7b finetuned on GSM8k. For more results, please refer to Appendix A. As a result of LoWR, all the channels have a larger norm product. With or without LoWR, the difference between $\|\mathbf{a}_i\| \cdot \|\mathbf{b}_i\|$ of different channels always reflects on $\|\mathbf{b}_i\|$, where the Pearson's R is close to 0.9. In Fig. 3b and Fig. 3c, we present the average $\|\mathbf{a}_i\| \cdot \|\mathbf{b}_i\|$ across layers and modules of a Llama-2-7b fine-tuned on GSM8k. Each bar has an error bar indicating the average standard deviation. Across all the layers, deep layers generally have a larger channel significance $\|\mathbf{a}_i\| \cdot \|\mathbf{b}_i\|$, indicating LoRA may

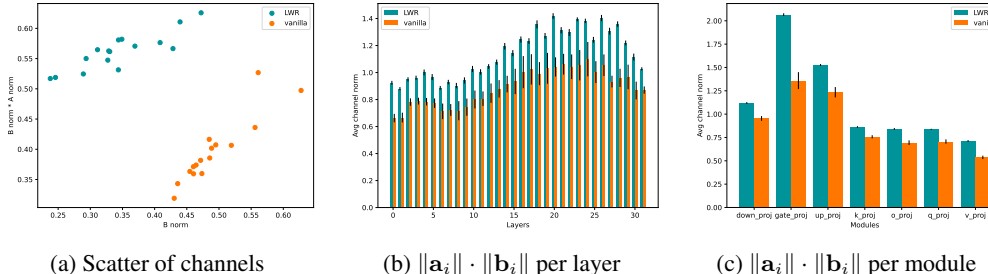

(a) Scatter of channels  (b) $\|\mathbf{a}_i\| \cdot \|\mathbf{b}_i\|$ per layer  (c) $\|\mathbf{a}_i\| \cdot \|\mathbf{b}_i\|$ per module

Figure 3: Results of channel significance ($\|\mathbf{a}_i\| \cdot \|\mathbf{b}_i\|$) of vanilla LoRA with and without LoWR for Llama-2-7b on GSM8k. **(a):** The scatter corresponds to the weight norm where each dot corresponds to a channel, the x-axis corresponds to $\|\mathbf{b}_i\|$ and the y-axis corresponds to $\|\mathbf{a}_i\| \cdot \|\mathbf{b}_i\|$. **(b):** The average $\|\mathbf{a}_i\| \cdot \|\mathbf{b}_i\|$ across different layers, each bar has an errorbar corresponding to the average standard deviation. **(c):** The average $\|\mathbf{a}_i\| \cdot \|\mathbf{b}_i\|$ across different modules, each bar has an errorbar corresponding to the average standard deviation.

have a greater effect on the deeper layers. Across all the modules, linear modules in the MLP have a larger channel significance than linear modules in the attention. The gate_proj receives the highest channel significance. Compared to vanilla LoRA, LoWR increases the average $\|\mathbf{a}_i\| \cdot \|\mathbf{b}_i\|$ while decreasing the average standard deviation on every layer and every module. The results show that LoWR successfully boosts the insignificant channels. By encouraging the update on insignificant channels, LoWR actively boosts performance by changing the training dynamic and introducing more effective rank utilization for LoRA modules.

### 4.2.3 CLOSER LOOK AT THE LOSS AND GRADIENT NORM

We present the gradient norm and loss in fine-tuning with or without LoWR for a closer look. Fig. 4 presents the loss and gradient norm of finetuning with Llama-2-7b on GSM8K. According to our analysis in Sec. 4.1, larger $\|\mathbf{a}_i\|$ leads to larger gradient $\nabla_{\mathbf{b}_i}\mathcal{L}$. With a larger gradient on $\mathbf{b}$, LoWR guides the LoWR module to further update the $\mathbf{b}$ for insignificant channels to record the information from the output gradient. Therefore, the gradient norm is generally larger with LoWR than with vanilla LoRA, especially at the late finetuning stage. Meanwhile, the loss is generally smaller with LoWR than with vanilla LoRA, indicating the efficacy of our LoWR in improving LoRA as a plug-in.

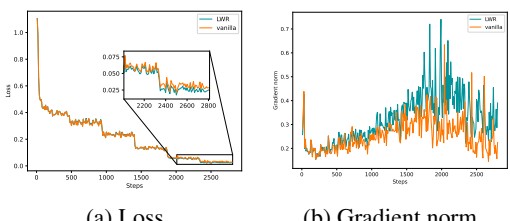

(a) Loss  (b) Gradient norm

Figure 4: The loss and gradient norm during the finetuning of Llama-2-7b on GSM8K. We report the results of vanilla LoRA and LoRA with LoWR in two colors. Generally, the loss is smaller and the gradient norm is larger with LoWR.

## 5 CONCLUSION

This paper presents a new perspective on low-rank adaptation (LoRA) and highlights the differences between LoRA and full-weight update. Unlike previous works that treat the two weight matrices as a weight decomposition, we show that the two weight matrices in LoRA serve as proxies for the input and output gradients during optimization. For a LoRA module of rank $r$, there are $r$ channels, each corresponding to two weight vectors as a set of input and output gradient proxies. While a branch of prior work proposes reducing insignificant channels for improved efficiency, we demonstrate that the difference between channels primarily stems from the weight vector in the weight matrix $\mathbf{B}$, which serves as a proxy for the output gradient. Furthermore, our analysis reveals that equivalent LoRA weights can have significantly different optimization trajectories during optimization. Based on our analyses, we provide a plug-in method, namely low-rank weight rescaling (LoWR), to stabilize the optimization of LoRA and guide LoRA to fully use the allocated rank (eliminating insignificant channels without reducing the rank). Experiments have verified the effectiveness of our proposed method. We hope that the insights and the proposed method provided in this paper can inform future works to investigate further and improve the efficacy of PEFT methods on LLMs.

## ETHICS STATEMENT

All the authors of this paper have read the ICLR Code of Ethics and will adhere to it during the submission process. This paper focuses on improving the performance of low-rank adaptation on large language models. This paper does not involve human subjects and poses no potentially harmful insights, methodologies, or applications.

## REPRODUCIBILITY STATEMENT

To ensure reproducibility, we provide details on the experimental setting of our experiments in the main text and the appendix. We provide detailed descriptions of how the experiments are conducted. We will provide the code upon acceptance of this paper.

## DECLARATION OF AI USE

We used Grammarly to check the grammar and fix typos in our writing. All ideas, analyses, and conclusions remain our own.

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

# A    ADDITIONAL RESULTS

## A.1    CHANNEL DIFFERENCE WITH LOWR

This section provides additional experiment results to demonstrate the channel difference and the comparison between vanilla LoRA and our LoWR. In Fig. 5 and Fig. 6, we show the results of channel norms with and without LoWR. Generally, the channel norm is consistently positively correlated with the norm of weight vector $\mathbf{b}$, demonstrating the difference between channels mainly reflects on the weight vector $\mathbf{b}$. With LoWR, $\|\mathbf{b}\|$ is generally decreased since the proposed method increase $\|\mathbf{a}\|$ and reduce $\|\mathbf{b}\|$. Meanwhile, the product of norm $\|\mathbf{a}\|\|\mathbf{b}\|$ is increased with LoWR, indicating the channels in the LoRA module finetuned with LoWR generally are more significant compared to the vanilla LoRA.

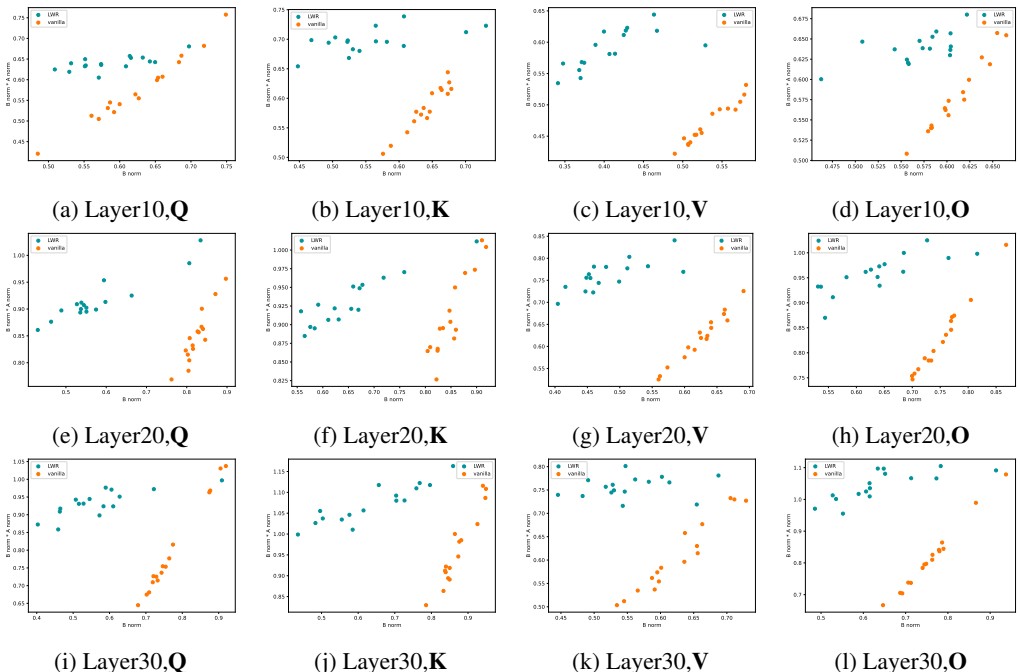

(a) Layer10,**Q**     (b) Layer10,**K**     (c) Layer10,**V**     (d) Layer10,**O**

(e) Layer20,**Q**     (f) Layer20,**K**     (g) Layer20,**V**     (h) Layer20,**O**

(i) Layer30,**Q**     (j) Layer30,**K**     (k) Layer30,**V**     (l) Layer30,**O**

Figure 5: The relationship between $\|\mathbf{a}_i\| \cdot \|\mathbf{b}_i\|$ and $\|\mathbf{b}_i\|$, with or without LoWR. The y-axis corresponds to $\|\mathbf{a}_i\| \cdot \|\mathbf{b}_i\|$ and the x-axis corresponds to $\|\mathbf{b}_i\|$. Each dot represents a channel in the LoRA Module. Vanilla LoRA and LoRA+LoWR are distinguished by two different colors. We present results across different layers of Llama2-7b finetuned on GSM8K.

## A.2    ANALYSIS ON THE HYPERPARAMETER SENSITIVITY OF LOWR

In Table 5, we provide results of finetuning DeBERTa-v3-base on MNLI in GLUE with different hyperparameters. For our proposed method, there are mainly two hyperparameters, threshold and ratio. Generally, the threshold determines the range of channels to have their weights rescaled, the ratio controls the rescaling strength, and the number of updated steps before each rescaling controls the rescaling frequency. Moderate hyperparameter setting typically achieves better performance. While the experiments in the main text rescale each channel every step, we further provide results of rescaling ever 5 steps, showing a possible way to adjust the rescaling frequency.

## A.3    CONNECTION AND COMPARISON BETWEEN LOWR AND LORA+

Hayou et al. [10] provides a brilliant work demonstrating that the learning rate for matrix $B$ should be larger than that for matrix $A$ for the gradient of both matrices to be of similar magnitude. From a different perspective, we show that weight matrix $B$ contains the output gradient information while $A$ contains the input information and reach a similar conclusion that weight matrix $B$ is more dif-

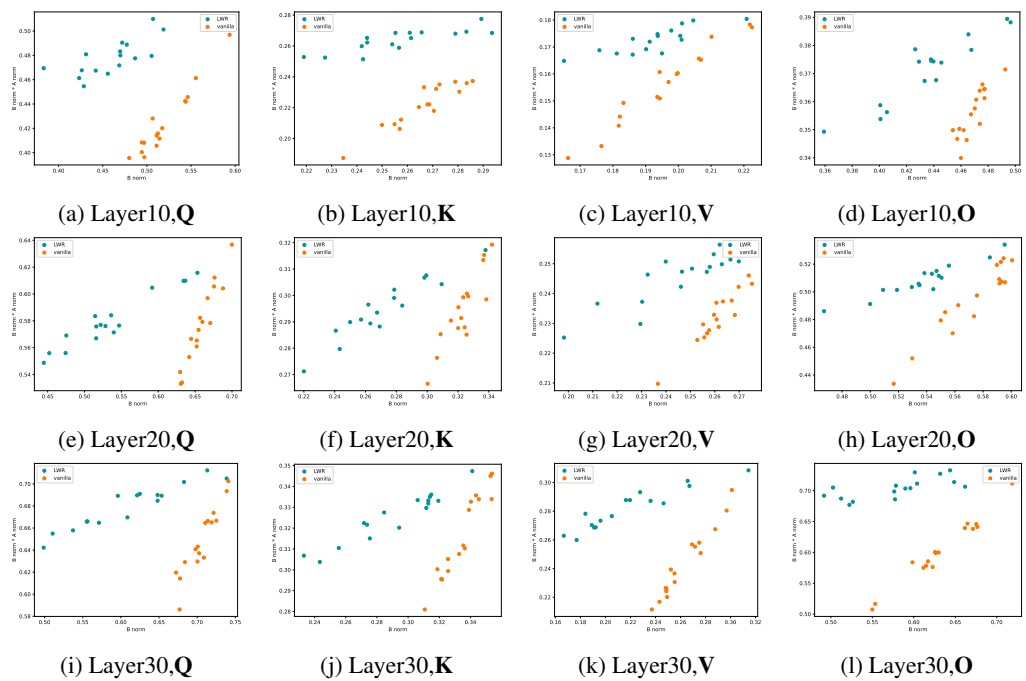

Figure 6: The relationship between $\|\mathbf{a}_i\| \cdot \|\mathbf{b}_i\|$ and $\|\mathbf{b}_i\|$, with or without LoWR. The y-axis corresponds to $\|\mathbf{a}_i\| \cdot \|\mathbf{b}_i\|$ and the x-axis corresponds to $\|\mathbf{b}_i\|$. Each dot represents a channel in the LoRA Module. Vanilla LoRA and LoRA+LoWR are distinguished by two different colors. We present results across different layers of Mistral-7B-v0.1 finetuned on GSM8K.

Table 5: The accuracy of finetuned DeBERTa-v3-base on MNLI in GLUE with LoWR with different hyperparameters.

| Threshold $\tau$ | ratio $\alpha$ | rescaling ever # of steps | accuracy (%) |
|---|---|---|---|
| $\infty$ | 0.001 | 1 | 83.00 |
| $\infty$ | 0.001 | 5 | **86.59** |
| 10 | 0.001 | 1 | 86.24 |
| 10 | 0.001 | 5 | 86.55 |
| 10 | 0.01 | 1 | 86.43 |
| 10 | 0.01 | 5 | 86.43 |
| 100 | 0.001 | 1 | 86.18 |
| 100 | 0.001 | 5 | 86.33 |
| 100 | 0.01 | 1 | 85.94 |
| 100 | 0.01 | 5 | 86.13 |
| baseline | - | - | 86.13 |

ficult to train. Generally, our weight rescaling method achieves a similar effect, encouraging the update on matrix $B$. The main difference is that our method is more fine-grained and proactively balance between channels during training. In Table 6, we provide results comparing our method with LoRA+.We use the official code provided in LoRA+ to conduct experiments. In the follow-

Table 6: Comparison between LoWR and LoRA+ finetuning DeBERTa-v3-base on MNLI in GLUE

| Method | Accuracy |
|---|---|
| LoRA+ | 86.19 |
| LoWR + LoRA+ | 86.59 |

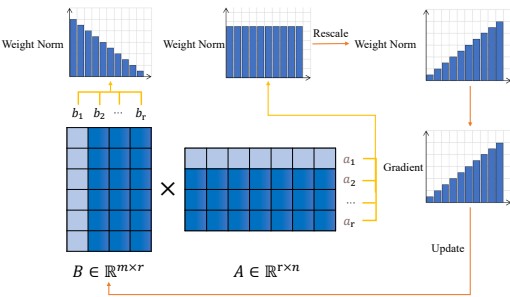

Figure 7: An illusration of the proposed method. As we theoretically analyzed and empirically validated, the imbalance mainly reflect on matrix $B$. To encourage the training of matrix $B$, we proposed to rescale between the matrix and increase the gradient norm on the matrix $B$.

ing, we report the results of finetuning Roberta-base on MNLI in GLUE. We follow the LoRA+ hyperparameter settings.

# B EXPERIMENTAL DETAILS

In this section, we provide details about our experiments. Generally, we conduct experiments following the setting in Li et al. [17]. The experiments are performed on a single A100. The source code of our experiments will be publicly available upon acceptance.

## B.1 EXPERIMENTAL DETAILS ON GSM8K

The setting for all the experiments on GSM8K is the same. The model is trained for 6 epochs with an initial learning rate set at $3e - 4$ and batch size set at 16. The learning rate scheduler type is "cosine". The hyper-parameter for LoWR is grid searched in a very small set: ($\tau \in \{+\infty, 10, 100\}$, $\alpha \in \{0.1, 1\}$).

## B.2 EXPERIMENTAL DETAILS ON GLUE

We use the default setting provided by [17] to conduct our experiments on GLUE. The hyper-parameter for LoWR is also grid searched in a very small set: ($\tau \in \{+\infty, 10, 100\}$, $\alpha \in \{0.0001, 0.001\}$). While the model is relatively sensitive to $\alpha$, we set it to a smaller value.

## B.3 A FIGURE TO ILLUSTRATE LOWR

In this section, we include a figure to further illustrate what the proposed LoWR do. In Fig. 7, we introduce an illustrative figure for our proposed method. To balance the channels and stabilize the training of the LoRA module, we propose to rescale between the two matrices. For a more detailed introduction, please refer to Sec. 4.1.

