# OpenReview forum: "LoWR: LoRA Weight Rescaling for Effective Rank Utilization beyond Reduction"
_ICLR.cc/2026/Conference — Submitted to ICLR 2026_

### Official Review · Reviewer_xs6N · 2025-10-16

**Soundness:** 3
**Presentation:** 3
**Contribution:** 2
**Rating:** 4
**Confidence:** 4

**Summary:**

This paper proposes a novel trick to boost LoRA fine-tuning called LoWR, which serves as a light-weight plug-in in the training process. The authors introduce a novel perspective that $A$ and $B$ can be viewed as proxies for the input and output gradients, respectively. Based on this, they studied the channel imbalance problem of the trainable matrices, which further motivates to dynamically rescale $A$ and $B$ for better robustness. Experiments have shown that LoWR can efficiently boost LoRA fine-tuning across various tasks.

**Strengths:**

1. The perspective to view $A$ and $B$ matrices as gradient proxies is fresh and novel, which has made it an interesting motivation for the algorithm design.

2. The proposed LoWR method is simple yet effective, as illustrated in its experimental results.

3. The paper provides detailed analysis to illustrate the effectiveness of LoWR, including an analysis regarding the norm product of channels and loss curves, which makes the method more convincing.

**Weaknesses:**

1. It seems that this paper does not discuss the computational overhead introduced by LoWR (e.g., monitoring weight norms, calculating variance, performing rescaling operations) and does not analyze its impact on training throughput, especially in distributed settings. Though the method seems a simple plug-in, it is hard for the reader to validate whether the performance gains come at the cost of training efficiency; the performance-efficiency trade-off remains unclear, and addressing this could largely enhance the completeness of this work.

2. The experimental section lacks direct comparisons with other important methods such as LoRA+. Although a combined experiment and discussion with LoRA+ is provided in the appendix, it is recommended to directly compare against it in the major experiments regarding 7B models.

3. The choice of key hyperparameter, such as the threshold $\tau$ and the ratio $\alpha$, lacks further explanation or ablation studies, making them appear somewhat heuristic. It is recommended to include some discussions on how to select this hyperparameters efficiently, and how their value could affect the final performance.

**Questions:**

1. Can the authors discuss LoWR's computational overhead, as well as the performance-efficiency trade-off?
2. In line 747 the authors claim that the LoWR is more fine-grained than LoRA+, which I find not clear enough. Can the authors give some further explanations on it?
3. Is LoWR sensitive to different batch sizes, learning rates, or hyperparameters $\tau$ and $\alpha$? I also wonder if LoWR can be more, less or the same effective when the model size scales up？

---

> ### Author Response · Authors · 2025-11-21
> **Rebuttal for Reviewer xs6N (part 1)**
>
> Dear Reviewer xs6N,
>
> Thank you for your time and valuable feedback regarding computational overhead. Below, we address each of your comments.
>
> > W1: It seems that this paper does not discuss the computational overhead introduced by LoWR (e.g., monitoring weight norms, calculating variance, performing rescaling operations) and does not analyze its impact on training throughput, especially in distributed settings. Though the method seems a simple plug-in, it is hard for the reader to validate whether the performance gains come at the cost of training efficiency; the performance-efficiency trade-off remains unclear, and addressing this could largely enhance the completeness of this work.
>
> > Q1: Can the authors discuss LoWR's computational overhead, as well as the performance-efficiency trade-off?
>
> Thank you for raising this important point. We will add a comprehensive complexity and overhead analysis to the paper.
>
> **The core operations in LoWR are lightweight**: Suppose the hidden size of the model is $n$ and the rank for a LoRA module is $r$, calculating the weight norm for each channel is of complexity $O(nr)$, and determining the ratio is of $O(r)$. Since the rank is typically much smaller than the hidden size, our method introduces generally a small computational overhead.
>
> In the following, we report the average time cost of each step when fine-tuning Deberta-v3-base on MNLI in GLUE with its std. We conduct our experiment using the official code from LoftQ and an A100 GPU. As shown in the table below, LoWR introduces only a ~6% time overhead on Deberta-v3-base and ~2% on Llama-7B, with no additional memory footprint.
>
> | Method       | Time (s)              |
> | ------------ | --------------------- |
> | LoftQ        | 0.06551$\pm$ 0.01275  |
> | LoftQ + LoWR | 0.06943$\pm$ 0.01074 |
>
> For larger models, we report the time cost of finetuning Llama-7B on MNLI for one epoch using an A100 GPU following the setting in LoRA+.
> | Method      | Time (s) |
> | ----------- | -------- |
> | LoRA        | 2.266    |
> | LoRA + LoWR | 2.313    |
>
> To further reduce the computational overhead, one can always choose to increase the number of steps between two reweighting operations. As demonstrated in Table 5, it barely affect the effectiveness of our method according to our experiments.
>
> > W2: The experimental section lacks direct comparisons with other important methods such as LoRA+. Although a combined experiment and discussion with LoRA+ is provided in the appendix, it is recommended to directly compare against it in the major experiments regarding 7B models.
>
> Thank you for the valuable suggestion. LoRA+ is indeed an important baseline. We provide experimental results and analysis in Appendix A.3. In the following, we provide additional experimental results following LoRA+:
>
> We follow the settings in LoRA+ and their open-source code to conduct our experiment. While the original experiment is conducted on various learning rate and LoRA+ ratio, we take the best setting in their experiments due to the limited time and computational resources.
>
> Roberta-base on GLUE
> | Method  | QQP | QNLI | SST2 |
> | ------  | --- | ---- | ---- |
> | LoRA+      | 89.10    |  92.33    |   93.57   |
> | LoRA+ + LoWR     | 89.29    |  92.46    |  94.15    |
>
> LLama 7B on MNLI
>
> | Method | MNLI |
> | ------ | ---- |
> | LoRA+  |   86.72   |
> | LoRA+ + LoWR       |  87.50    |
>
> The key insight we wish to highlight is that LoWR is **orthogonal and complementary** to LoRA+, as well as to other LoRA variants. Our method addresses a different aspect of the optimization problem—channel-level imbalance—while LoRA+ operates at the matrix-level.
>
> In the paper, we combine our method with various different LoRA variants (e.g. LoftQ, AdaLoRA, PiSSA, OLoRA, etc.) to validate the effectiveness of our proposed method. The fact that LoWR can be seamlessly combined with LoRA+ (and other methods like AdaLoRA, PiSSA, etc.) to yield further performance improvements is a strong testament to its unique contribution and general applicability.

---

> > ### Author Response · Authors · 2025-11-21
> > **Rebuttal for Reviewer xs6N (part 2)**
> >
> > > W3: The choice of key hyperparameter, such as the threshold $\tau$ and the ratio $\alpha$, lacks further explanation or ablation studies, making them appear somewhat heuristic. It is recommended to include some discussions on how to select this hyperparameters efficiently, and how their value could affect the final performance.
> >
> > We appreciate the comment on hyperparameters. Their roles are intuitive: $\alpha$ controls the rescaling strength, and $\tau$ acts as a stability safeguard. The hyperparameter $\alpha$ determines the scaling strength, larger $\alpha$ means the scaling ratio is more deviated from $1$. Since our method enlarges $A$, the threshold $\tau$ is a save-proof to prevent matrix $A$ from being too much larger than matrix $B$. For a channel with $\|a_i\|/\|b_i\| >\tau$, the rescaling woud not be carried out. Generally, smaller $\tau$ and $\alpha$ means the behaviour with our proposed method is closer to vanilla LoRA.
> >
> > In Appendix A.2 (line 688-695), we present the analysis on the hyperparameter with experiments to demonstrate the hyperparameter sensitivity of our method. Moderate hyperparameter setting typically achieves better performance.
> >
> >
> > > Q2: In line 747 the authors claim that the LoWR is more fine-grained than LoRA+, which I find not clear enough. Can the authors give some further explanations on it?
> >
> > Thank you for the opportunity to clarify. The term "fine-grained" refers to the level of granularity at which our method operates:
> >
> > * LoRA+ applies different learning rates to the entire A and B matrices (matrix-level).
> >
> > * Our method (LoWR) performs rescaling operations for each individual channel within those matrices (channel-level).
> >
> > Since a channel is a constituent part of a matrix, our intervention is indeed more fine-grained. We would like to further explain the difference between our method and LoRA+ from two perspectives:
> >
> > * Different motivation: In this paper, we focus on balancing the channels in a LoRA module (each corresponds to a pair of row/column vector in the two matices A and B), while LoRA+ focus on the coordinates the update of the two matrices A and B.
> >
> > * Different approach: Since we have different motivation, we apply rescaling on each channel instead of adjusting learning rate for the whole matrix. Since our method is channel-wise, therefore, we claim that the proposed method is more fine-grained than LoRA+.
> >
> > Besides the difference in methods, in this paper, we provide a new perspective to analyze LoRA and demonstrate that the matrix $B$ corresponds to output gradient , which is harder to learn. We hope the provided insights can innovate furture works.
> >
> > > Q3: Is LoWR sensitive to different batch sizes, learning rates, or hyperparameters $\tau$ and $\alpha$? I also wonder if LoWR can be more, less or the same effective when the model size scales up？
> >
> > In this paper, we incooperate our method with various different LoRA variants (e.g. LoftQ, AdaLoRA, PiSSA, OLoRA, etc.) trying to test the robustness of our method in different scenarios.
> > We also provide an analysis and experimental results on the hyperparameter sensitivity in Appendix. A.2. Generally, our method is robust to hyperparameter settings. However, smaller batch sizes, larger learning rates, these factors can lead to unstable training, which may require a more cautious hyperparameter setting (smaller threshold $\tau$ and ratio $\alpha$).
> >
> > As for the model size, the motivation of the proposed method does not pose any assumptions on the model size. While we follow the settings from previous works to conduct our experiments, it is reasonable to assume that the model size would not affect the effectiveness of our method.
> >
> > We hope our rebuttal can address your concerns and we are more than happy to address any further comments. Thanks again for the valuable feedback.

---

> > > ### Comment · Reviewer_xs6N · 2025-11-23
> > > **Thanks for the response.**
> > >
> > > Thanks for the detailed response, which has successfully addressed most of my concerns. I'll raise the rating accordingly.

---

> > > > ### Author Response · Authors · 2025-11-24
> > > >
> > > > Dear Reviewer xs6N,
> > > >
> > > > We are glad that our rebuttal has addressed your concerns. We sincerely appreciate your effort devoted to the reviewing process. Thank you, and we will always be available for any further discussions.
> > > >
> > > > Best regards,
> > > >
> > > > Authors

---

### Official Review · Reviewer_hQ5R · 2025-10-26

**Soundness:** 2
**Presentation:** 2
**Contribution:** 2
**Rating:** 4
**Confidence:** 3

**Summary:**

The paper proposes a new algorithm for optimizing LoRA-style PEFT methods. First, the paper makes an observation that if we write the adapter product $BA = \sum_i b_i a_i^\top$, while $|| a_i||$ acts as a proxy of the input to the layer and is stable,  $|| b_i||$ is related to the gradient of the output of the layer and can fluctuate more. The paper proposes that an optimization algorithm should make smaller steps for $a_i$ and bigger steps for $b_i$, and in order to do so, by writing down the gradient with respect to $a_i, b_i$, the ratio $||a_i||/||b_i||$ should remain large during the training. For this reason, the proposed algorithm should rescale $a_i, b_i$ if  the ratio $||a_i||/||b_i||$  is small. In experiment, the paper shows that the algorithm can be combined with existing PEFT methods and improves their performance on common benchmarks.

**Strengths:**

- The paper made an interesting observation that we should distinguish between the channels during the training process and provides empirical evidence for this observation. This observation goes beyond the prior work  (LoRA+) that says the two matrices A and B play different roles and require different learning rate.
- The proposed algorithm is also generic to all LoRA-style methods and can improve most existing methods.

**Weaknesses:**

- The paper lacks clarity on some of the key points (see questions below).
- Statements in this paper lacks rigor. Specifically, Proposition 3.1 and Proposition 4.1 both lack rigor of a mathematical proposition. I suggest that the authors call Proposition 3.1 an observation.
- While the experiments show some improvement, the proposed algorithm is purely based on empirical observation and lacks mathematical justifications.
- The experiments seem sparse and lack some of the baselines. More specifically, LoRA+ should be a direct comparable algorithm because it is also an optimization based approach. Some of the baselines in Table 3 are really low. In this table, LoRA should also be added.
- There are a lot of typos, such as a lot of inconsistent use of bold and normal symbols, or in the proof of Proposition  4.1.

**Questions:**

- In figure 2, what exactly do the dots represent? Are these pairs of $(||b_i||, ||a_i||||b_i||)$ collected across all layers and all values of $i$?
- I don't understand why in Algorithm 1, why don't we just scale so that $ ||a_i||/||b_i|| = \tau$? Any reasons for the choice in the algorithm?

---

> ### Author Response · Authors · 2025-11-21
> **Rebuttal for Reviewer hQ5R (part 1)**
>
> Dear reviewer hQ5R,
>
> We appreciate the time you devoted to the reviewing process are deeply grateful for your meticulous review. Our point-by-point responses are detailed below.
>
> > W1: The paper lacks clarity on some of the key points (see questions below).
> > * Statements in this paper lacks rigor. Specifically, Proposition 3.1 and Proposition 4.1 both lack rigor of a mathematical proposition. I suggest that the authors call Proposition 3.1 an observation.
> > * There are a lot of typos, such as a lot of inconsistent use of bold and normal symbols, or in the proof of Proposition 4.1.
>
> We sincerely thank the reviewer for pointing out these issues regarding mathematical rigor and typos. We have conducted a line-by-line review to unify all mathematical notation (e.g., bold vs. normal symbols) and correct all typos. We will keep improving our paper. We apologize for these oversights in the previous version and appreciate the reviewer's meticulousness in helping us improve the paper's quality.
>
> > W2: While the experiments show some improvement, the proposed algorithm is purely based on empirical observation and lacks mathematical justifications.
>
> Our method is not merely an empirical design but is directly motivated by a sequence of theoretical insights and empirical validations that form a coherent justification: We would like to take this opportunity to  summarize and emphasize the insights that guides the design of our algorithm:
> * **Theoretical analysis(Sec. 3.1, 3.2)：** **We theoretically provide a brand new perspective to analyze LoRA, showing that two matrices in LoRA learns the input and output gradient respectively.**
> * **Theoretically-Grounded Hypothesis**： Based on previous theoretical works that showing gradients could drastically change, we hypothesize that the matrix corresponding to output gradient is more difficult to train.
> * **Empirical Validation (Sec. 3.3)**： To validate the hypothesis, we provide empirical evidence in Sec. 3.3 such that the weight norm difference between different channels of a LoRA module mainly reflects on the matrix corresponding to output gradient (namely LoRA B matrix).
>
> The proposed LoWR algorithm is the natural and principled outcome of this analysis. It directly addresses the identified problem of channel imbalance in B, which is rooted in our theoretical framework.

---

> ### Author Response · Authors · 2025-11-21
> **Rebuttal for Reviewer hQ5R (part 2)**
>
> > W3: The experiments seem sparse and lack some of the baselines. More specifically, LoRA+ should be a direct comparable algorithm because it is also an optimization based approach. Some of the baselines in Table 3 are really low. In this table, LoRA should also be added.
>
> **LoRA+ baseline**：
> We agree that LoRA+ is an important baseline. In Appendix, we provide experimental results , showing that our method, as a plug-in, is orthogonal to LoRA+ and could be used to further boost the performance of LoRA+. Here we provide additional experimental results. We follow the settings in LoRA+ and their open-source code to conduct our experiment. While the original experiment is conducted on various learning rate and LoRA+ ratio, we take the best setting in their experiments due to the limited time and computational resources.
>
> Roberta-base on GLUE
> | Method  | QQP | QNLI | SST2 |
> | ------  | --- | ---- | ---- |
> | LoRA+      | 89.10    |  92.33    |   93.57   |
> | LoRA+ + LoWR     | 89.29    |  92.46    |  94.15    |
>
> LLama 7B on MNLI
>
> | Method | MNLI |
> | ------ | ---- |
> | LoRA+  |   86.72   |
> | LoRA+ + LoWR       |  87.50    |
>
>
> To further highlight the difference between our method and LoRA+, our method focus on balancing between channels of a LoRA module, while LoRA+ focus on adjusting the learning rate for A and B, the two matrices in LoRA. In Appendix A.3, we provide an analysis on the connection and difference between our method and LoRA+.
>
> **Table 3 and Low Baselines:**
> Table 3 specifically evaluates methods under **extreme 2-bit quantization** based on the LoftQ framework, which explains the lower absolute performance, as this is a profoundly challenging setting. We follow the setting in [1] to conduct our experiments, where each parameter is quantized to 2 bit, which is the reason for low baselines. According to the results in [1], vanilla LoRA achieves a similar performance to Full fine-tuning. To provide context, we quote the results of Vanilla LoRA (without quantization) from [1]:
>
> | **Rank** | **Method** | **MNLI** | **QNLI** | **RTE** | **SST** | **MRPC** | **CoLA** | **QQP** | **STSB** | **SQuAD** | **ANLI** |
> |----------|------------|----------|----------|---------|---------|----------|----------|---------|----------|-----------|----------|
> |         |           | m / mm   | Acc      | Acc     | Acc     | Acc      | Matt      | Acc     | P/S Corr | EM/F1     | Acc      |
> |         | Full FT    | 90.5/90.6 | 94.0     | 82.0    | 95.3    | 89.5/93.3 | 69.2     | 92.4/89.8 | 91.6/91.1 | 88.5/92.8 | 59.8     |
> | 16       | LoRA       | 90.4/90.5 | 94.6     | 85.1    | 95.1    | 89.9/93.6 | 69.9     | 92.0/89.4 | 91.7/91.1 | 87.3/93.1 | 60.2     |
>
> This clearly shows that in the standard full-precision setting, LoRA performs comparably to full fine-tuning, while our method provides significant gains in the low-bit regime we are studying.
>
> > Q1: In figure 2, what exactly do the dots represent? Are these pairs of $\|b_i\|$, $\|a_i\|\|b_i\|$ collected across all layers and all values of $i$ ?
>
> The understanding is generally correct. As we described under Fig. 2 (line 237-241), each dot corresponds to a channel of LoRA modules in the first layer of Llama2-7B and Mistral-7B, corresponding to pairs of $\|b_i\|$, $\|a_i\|\|b_i\|$ collected across all values of $i$. In Fig. 2, we want to demonstrate that the imbalance between channels mainly reflects on matrix $B$, which corresponds to learning the output gradient.
>
> > Q2: I don't understand why in Algorithm 1, why don't we just scale so that $\|a\|/\|b\|=\tau$? Any reasons for the choice in the algorithm?
>
> This is an excellent question. Abruptly rescaling to a fixed ratio $\tau$ can be overly aggressive and disrupt the model's optimization trajectory. Our design employs a soft, incremental adjustment towards a more balanced state. The principle behind this design is to keep the training stable, analogous to the use of gradient clipping or learning rate warmup for example.
>
> Thank you again for your valuable feedback. We will keep polishing our paper. We hope our rebuttal can address your concerns, and we look forward to your further feedback. We are more than happy to address any further comments.
>
> [1] Yixiao Li, Yifan Yu, Chen Liang, Pengcheng He, Nikos Karampatziakis, Weizhu Chen, and Tuo Zhao. Loftq: Lora-fine-tuning-aware quantization for large language models.

---

> > ### Comment · Reviewer_hQ5R · 2025-11-27
> >
> > Thank you for your response. Below are my follow-up comments.
> >
> > - Regarding the experiments: The current experiment setup of the paper tries different baselines for different datasets (LoftQ for GLUE, LoRA/oLoRA for GSM8K, AdaLoRA for Squad), which makes me unsure how to evaluate the results. It is necessary to provide the experiment results when the baseline is LoRA (or using the same baselines across datasets) on standard benchmarks like GLUE, commonsense, etc. While in your response, you mentioned that "LoRA performs comparably to full fine-tuning", it is necessary to see the performance of LoWR with it, and have an analysis, for example, as LoRA+ did.
> > - The new hyperparameters $\tau, \alpha$: From the newly provided results in the appendix, it is still unclear how to set/tune these parameters ($\tau$ gives the best results?).
> > - Regarding the claim on the theoretical analysis: As I mentioned in my review, Proposition 3.1 is just an observation. The paper hinges on this observation and doesn't provide justifications for the benefits of it. For comparison, LoRA+ can show the benefit of separating the learning rates for the stability and efficiency of the training (with proper definitions).
> >
> > Overall, I think while the new perspective in this paper is interesting, it needs more clarity and results, both theoretical and empirical.

---

> > > ### Author Response · Authors · 2025-12-02
> > > **Reply to further comments from Reviewer hQ5R**
> > >
> > > We appreciate the further comments from Reviewer hQ5R. In the following, we provide further clarification on the comments.
> > >
> > > * **Reply to comments on the experiment**: Our proposed method works as a plug-in to apply to LoRA and its variants; therefore, the effectiveness of our method is evaluated by comparing the performance of a LoRA variant with or without our method. To fully verify the effectiveness of our method, we apply our method to **vanilla LoRA** (on GSM8K), **initialization modified variants** (PiSSA, OLoRA on GSM8K), **AdaLoRA** that adaptively adjusts LoRA during training, **LoftQ** focusing on quantization, and **LoRA+** adjusting the learning rate of the two weight matrices. With these various LoRA variants, we follow their experimental setting to conduct the experiments. The results show that our method is effective in boosting the performance of vanilla LoRA and various variants of LoRA.
> > >
> > > * **Reply to the hyper-parameter setting**: According to our sensitivity analysis in Appendix A.2, the proposed method is generally robust to different hyperparameters. For stable performance, we generally set the hyperparameters $\tau$ and $\alpha$ to a small value (for example, $\tau=10$, $\alpha=0.01$), which can restrict our proposed method close to vanilla LoRA and keep the training procedure stable.
> > >
> > > * **Reply to the comments on our theoretical analyses**: In proposition 3.1, we provide a new perspective to understand LoRA (one matrix learning the input and one matrix learning the output gradient). Our core motivation is grounded in established theoretical work showing that output gradients can exhibit high variance and change abruptly, especially near convergence when the model may oscillate around a minimum [1, 2, 3]. This motivation is further justified by the empirical evidence we provide in Sec.  3.3.
> > >
> > > We hope our response clarifies the motivation of our proposed method and the evaluation results provided in the paper.
> > >
> > > [1] Jeremy M Cohen, Simran Kaur, Yuanzhi Li, J Zico Kolter, and Ameet Talwalkar. Gradient descent on neural networks typically occurs at the edge of stability.
> > >
> > > [2] Alex Damian, Eshaan Nichani, and Jason D Lee. Self-stabilization: The implicit bias of gradient descent at the edge of stability.
> > >
> > > [3] Fartash Faghri, David Duvenaud, David J Fleet, and Jimmy Ba. A study of gradient variance in deep learning.

---

### Official Review · Reviewer_Nnks · 2025-11-04

**Soundness:** 3
**Presentation:** 2
**Contribution:** 3
**Rating:** 6
**Confidence:** 3

**Summary:**

The paper provides a new perspective for LoRA, where each channel in LoRA corresponds to a pair of $a_i$ acting as a proxy of the input and $b_i$ acting as a proxy of the output gradient. Basically, the paper decomposes the gradient updates of W=AB into two components, with one being less stable than the other, and suggests that we can rebalance them to make them more stable while not changing the equivalent weight update W.
Based on the analysis from this new perspective, the authors provide a simple method to scale up the component $a_i$ and scale down $b_i$ in the LoRA weight update since the output gradient is usually noisier. The rescaling only happens if the norm ratio falls under some threshold. The scaling factor is determined such as it increases a for channel with small b and for channels that are more imbalance.

**Strengths:**

- The method is simple and considered itself a plug-in that can be applied to LoRA, variants of LoRA, and also adaptive rank allocation methods
- The results are strong across the board
- The authors propose theoretical insights into why the rescaling is needed by interpreting the gradient update as the sum of two components in the direction of the input and the output gradient.

**Weaknesses:**

- The method introduces several hyper-parameter that's not intuitive to tune: threshold $\tau$ for when to rescale, $\alpha$ for the rescaling factor. Since the method acts as a plugin to some base LoRA method, the base method can already be expensive to run, which makes hyper parameter tuning harder to run even on a small set.
- The main motivation for the method is the input for each layer is less "noisy" than the output gradient. Can this noise be quantified empirically or theoretically.

**Questions:**

- What were used as the validation set when tuning $\tau$ and $\alpha$ for each experiment?
- How transferable are $\tau$ and $\alpha$. Do we need to tune per architecture or also per dataset?

---

> ### Author Response · Authors · 2025-11-21
> **Rebuttal for Reviewer Nnks**
>
> Dear Reviewer Nnks,
>
> We greatly appreciate your insightful comments. We are glad that you find our work simple but effective. We address each of your comments in the following.
>
> > W1: The method introduces several hyper-parameter that's not intuitive to tune: threshold $\tau$ for when to rescale, $\alpha$ for the rescaling factor. Since the method acts as a plugin to some base LoRA method, the base method can already be expensive to run, which makes hyper parameter tuning harder to run even on a small set.
>
> We thank the reviewer for this important point regarding hyperparameter tuning.  The proposed method mainly envolve two hyperparameters. The hyperparameter $\alpha$ determines the scaling strength, larger $\alpha$ means the scaling ratio is more deviated from $1$. Since our method enlarges $A$, the threshold $\tau$ is a save-proof to prevent matrix $A$ from being too much larger than matrix $B$. For a channel with $\|a_i\|/\|b_i\| >\tau$, the rescaling woud not be carried out. Generally, smaller $\tau$ and $\alpha$ means the behaviour with our proposed method is closer to vanilla LoRA.
>
> We provide a sensitivity analysis in Appendix A.2. Generally, moderate hyperparameter setting typically achieves better performance (e.g. $\tau=10$, $\alpha=0.01$).
>
> > W2: The main motivation for the method is the input for each layer is less "noisy" than the output gradient. Can this noise be quantified empirically or theoretically.
>
> The term "noisy" might be an oversimplification. Our core motivation is grounded in established theoretical work showing that output gradients can exhibit high variance and change abruptly, especially near convergence when the model may oscillate around a minimum [1, 2, 3].
>
> > Q1: What were used as the validation set when tuning $\tau$ and $\alpha$ for each experiment?
>
> For experiments, we  align our experimental setting with the experimental settings of the baseline methods we compared against. Since many of these tasks do not have a standard validation set, we used training loss (not accuracy) as a proxy for stability and convergence behavior.
>
> > Q2: How transferable are $\tau$ and $\alpha$. Do we need to tune per architecture or also per dataset?
>
> According to our sensitivity analysis in Appendix A.2, the proposed method is generally robust to different hyperparameters. However, it do need a adjustment accross different architectures and datasets. For smaller model and dataset, we tend to set a smaller $\tau$ and $\alpha$, which can restrict our proposed method close to vanilla LoRA and keep the training procedure stable.
>
> We hope our response can address your concerns, and we look forward to receiving your further feedback. We are more than happy to answer any further comments.
>
> [1] Jeremy M Cohen, Simran Kaur, Yuanzhi Li, J Zico Kolter, and Ameet Talwalkar. Gradient descent on neural networks typically occurs at the edge of stability.
>
> [2] Alex Damian, Eshaan Nichani, and Jason D Lee. Self-stabilization: The implicit bias of gradient descent at the edge of stability.
>
> [3] Fartash Faghri, David Duvenaud, David J Fleet, and Jimmy Ba. A study of gradient variance in deep learning.

---

### Official Review · Reviewer_5p8G · 2025-11-10

**Soundness:** 3
**Presentation:** 2
**Contribution:** 2
**Rating:** 4
**Confidence:** 4

**Summary:**

To enhance the parameter-efficient fine-tuning (PEFT) of large language models (LLMs), the authors propose a principled plug-in method that guides LoRA modules to fully utilize their allocated rank. Unlike prior approaches that dynamically prune insignificant channels, this method reweights the two LoRA weight matrices based on their theoretical roles as proxies for input and output gradients, thereby stabilizing training and improving channel utilization. Experiments on popular LLMs (e.g., LLaMA, Mistral) and benchmarks (e.g., GSM8K, GLUE, SQuAD) demonstrate that the proposed method consistently boosts the performance of LoRA and its variants as a simple yet effective plug-in.

**Strengths:**

* S1: This paper interprets the two matrices in LoRA as “input proxy” and “output gradient proxy”, a novel and well-founded perspective supported by both mathematical analysis and empirical validation.
* S2: The proposed method, LoWR, does not alter the base LoRA architecture but rescales channels during training, making it easy to implement and highly compatible with various LoRA variants.
* S3: Extensive experiments across multiple models and tasks verify that the proposed method consistently delivers performance gains in diverse scenarios.

**Weaknesses:**

* W1: Equation (12) reflects an engineering-oriented and intuitive design but lacks rigorous theoretical derivation or boundary analysis, as well as sensitivity analysis on $\gamma$ or comparisons with alternative formulations.
* W2: The paper does not analyze or discuss the computational complexity or overhead of LoWR. Although LoWR involves simple scaling operations, performing rescaling at every step across all layers and channels may introduce additional costs.
* W3: The overall presentation needs improvement — adding one or two illustrative figures in Section 3 (the methodology part) would enhance clarity and readability.

**Questions:**

* Q1: Why is the computation of $\gamma$ designed as a linear difference form $(\max|b| - |b_i|)$ instead of a relative ratio or normalized formulation?
* Q2: Are there alternative designs for $\gamma$? Have any comparative experiments been conducted to analyze their effects?
* Q3: How does LoWR’s time overhead and memory consumption compare to baseline methods across different model scales (e.g., 7B, 13B)?
* Q4: In the methodology section, a figure could be added to illustrate why changing only the scale without altering the product affects training dynamics.

---

> ### Author Response · Authors · 2025-11-21
> **Rebuttal for Reviewer 5p8G**
>
> Dear Reviewer 5p8G，
>
> We sincerely thank you for your thorough review and valuable feedback. In the following, we address each of your comments.
>
> > W1: Equation (12) reflects an engineering-oriented and intuitive design but lacks rigorous theoretical derivation or boundary analysis, as well as sensitivity analysis on $\gamma$ or comparisons with alternative formulations.
>
> The motivation of our method is firmly grounded in solid theoretical analysis and empirical observations:
> * **We theoretically provide a brand new perspective to analyze LoRA, showing that the two matrices in LoRA learns the input and output gradient, respectively.**(Sec. 3.1, 3.2)
> * Based on previous theoretical works that show gradients could drastically change, we hypothesize that the matrix corresponding to the output gradient is more difficult to train.
> * To validate the hypothesis, we provide empirical evidence in Sec. 3.3 such that the weight norm difference between different channels of a LoRA module mainly reflects on the matrix corresponding to output gradient (LoRA B).
>
> Based on the above theoretical analysis and empirical observations, we propose an algorithm to balance between channels of a LoRA module and improve its performance.
>
> Sensitivity analysis: We have provided results in Table 5 in Appendix to show that our proposed method is robust to various hyperparameter settings.
>
> Alternative formulations: Our primary contribution lies in providing new insights and proposing a new perspective for understanding LoRA. We believe this opens avenues for future work to explore other formulations based on our insights.
>
> > W2: The paper does not analyze or discuss the computational complexity or overhead of LoWR. Although LoWR involves simple scaling operations, performing rescaling at every step across all layers and channels may introduce additional costs.
>
> > Q3: How does LoWR’s time overhead and memory consumption compare to baseline methods across different model scales (e.g., 7B, 13B)?
>
> We will add a complexity analysis to the paper in the Appendix. Our proposed method LoWR does not introduce additional memory consumption and introduce a little time overhead to compute the rescaling scale. Suppose the hidden size is $n$ and the rank is $r$, computing the channel norm is of complexity $O(nr)$ and determine the rescaling ratio requires $O(r)$.
> While the rank of LoRA modules is generally much smaller than the hidden size, the overhead of our method is much smaller compared to the forward process of complexity $O(n^2)$. In the following, we report the average time cost of each step when fine-tuning Deberta-v3-base on MNLI in GLUE with its std. We conduct our experiment using the official code from LoftQ and an A100 GPU.
>
> | Method       | Time (s)              |
> | ------------ | --------------------- |
> | LoftQ        | 0.06551$\pm$ 0.01275  |
> | LoftQ + LoWR | 0.06943$\pm$ 0.01074 |
>
> For larger models, we report the time cost of finetuning Llama-7B on MNLI for one epoch using an A100 GPU following the setting in LoRA+.
> | Method      | Time (s) |
> | ----------- | -------- |
> | LoRA        | 2.266    |
> | LoRA + LoWR | 2.313    |
>
> Our method as a plug-in will increase a little computational overhead (in this case, approximately 6% on Deberta-v3-base and 2% on Llama-7B). To further reduce the computational overhead, one can always choose to increase the number of steps between two reweighting operations. As demonstrated in Table 5, it barely affects the effectiveness of our method according to our experiments.
>
> > W3: The overall presentation needs improvement — adding one or two illustrative figures in Section 3 (the methodology part) would enhance clarity and readability.
>
> > Q4: In the methodology section, a figure could be added to illustrate why changing only the scale without altering the product affects training dynamics.
>
> Thank you for the suggestion. We have added a figure in the Appendix (Fig. 7) that visually illustrates how rescaling influences training dynamics without altering the product of the LoRA matrices. This figure helps clarify the motivation behind our approach and enhances the readability of Section 3.
>
> > Q1: Why is the computation of $\gamma$ designed as a linear difference form instead of a relative ratio or normalized formulation?
>
> > Q2: Are there alternative designs for $\gamma$? Have any comparative experiments been conducted to analyze their effects?
>
> We chose the current design for its simplicity and direct alignment with our theoretical insight: it effectively captures the channel-wise imbalance without introducing unnecessary complexity. While alternative designs are possible, our goal was to validate whether a principled rescaling mechanism grounded in our analysis can improve LoRA. In Table 5, we provide experimental results to demonstrate the robustness of the simple design to hyperparameter settings.
>
> Thank you again for your insightful suggestions. We are more than happy to address any further comments.

---

### Meta-Review · Area_Chair_93TG · 2026-01-07

**Summary:**

This paper presents a LoRA-based method for fine-tuning large models.

The reviewers were mainly concerned about the following problems.
* The algorithm is heuristic, and its theoretical analysis is insufficient.
* The hyper-parameters were not intuitive to tune and the robustness was not validated.
* The efficiency of the algorithm was not reported.
* Missing comparison to stronger methods like LoRA+.
* The paper needs further revision (statements were unclear).

**Reviewer Concerns:**

The authors addressed part of the concerns, but the method was not yet established on a solid theoretical analysis.

**Reviewer Scores:**

I think the three negative reviewers will keep their scores unchanged. The overall rating is not sufficient for the paper to get accepted.

---

### Decision · Program_Chairs · 2026-01-26

Reject